# Diagnostic Accuracy of 128-Slice Single-Source CT for the Detection of Dislocated Bucket Handle Meniscal Tears in the Setting of an Acute Knee Trauma—Correlation with MRI and Arthroscopy

**DOI:** 10.3390/diagnostics13071295

**Published:** 2023-03-30

**Authors:** Georg Gohla, Mareen Sarah Kraus, Isabell Peyker, Fabian Springer, Gabriel Keller

**Affiliations:** 1Department of Diagnostic and Interventional Neuroradiology, University-Hospital of Tuebingen, 72076 Tuebingen, Germany; 2Department of Diagnostic and Interventional Radiology, University-Hospital of Tuebingen, 72076 Tuebingen, Germany; 3Department of Diagnostic Radiology, IWK Health Care Centre, 5850 University Avenue, Halifax, NS B3K 6R8, Canada

**Keywords:** bucket handle tear, emergency radiology, computed tomography, magnetic resonance imaging, arthroscopy

## Abstract

(1) Background: Meniscal tears are amongst the most common knee injuries. Dislocated bucket handle meniscal tears in particular should receive early intervention. The purpose of this study was to evaluate the diagnostic performance of CT in detecting dislocated bucket handle meniscal tears compared with the gold-standard MRI and arthroscopy. (2) Methods: Retrospectively, 96 consecutive patients underwent clinically indicated CT of the knee for suspected acute traumatic knee injuries (standard study protocol, 120 kV, 90 mAs). Inclusion criteria were the absence of an acute fracture on CT and a timely MRI (<6 months). Corresponding arthroscopy was assessed. Two experienced musculoskeletal radiologists analyzed the images for dislocated bucket handle meniscal tears, associated signs thereof (double posterior cruciate ligament sign, double delta sign, disproportional posterior horn sign), and subjective diagnostic confidence on a 5-point-Likert scale (1 = ‘non-diagnostic image quality’, 5 = ‘very confident’). (3) Results: Dislocated bucket handle meniscal tears were detected on CT by standard three-plane bone kernel reconstructions with a sensitivity of 90.7% and a specificity of 99.3% by transferring the knowledge of established MRI signs. The additional use of soft-tissue kernel reconstructions in three planes increased the sensitivity by 4.0% to 94.7%, specificity to 100%, inter-rater agreement to 1.0, and the diagnostic confidence of both readers improved to a median 4/5 (‘confident’) in both readers. (4) Conclusions: Trauma CT scan of the knee with three-plane soft-tissue reconstructions delivers the potential for the detection of dislocated bucket handle meniscal tears with very high diagnostic accuracy.

## 1. Introduction

Acute knee trauma is one of the most common conditions observed in surgical and orthopedic emergency departments. Within the knee joint, meniscal injuries are the second most common, with an incidence of 12–14% and a prevalence of 61 cases per 100,000 persons [1]. Among meniscal injuries, bucket handle meniscal tears are a particularly severe form, and the frequency has been reported between 9 and 26% [2,3]. Early diagnosis with a contemporary surgical repair is crucial for the outcome of bucket handle meniscal tears, thereby significantly impacting health-related quality of life [4].

When patients present with acute knee trauma, a thorough medical history and physical examination are performed to identify the cause and location of the injury. Then, plain radiographs are obtained to evaluate the possibility of bone fractures. However, radiographic findings may be non-conclusive and further imaging needed to assess the soft tissue structures of the knee joint. For this reason, computed tomography (CT) scans are often performed to evaluate the morphology of the fracture, while magnetic resonance imaging (MRI) is the gold standard for detecting soft tissue injuries, especially for non-displaced ligamentous or meniscal damages that are undetectable on CT [5]. In contrast to MRI, CT is often the first cross-sectional examination in patients with acute knee injuries as they are widely available, have few contraindications, and provide quick, time-saving examinations. In recent decades, technological advancements have improved the resolution and quality of CT images, providing higher spatial resolution, better soft tissue contrast, and improved signal-to-noise ratio (SNR) [6]. As a result, CT scans have become an increasingly valuable tool for evaluating a variety of acute knee injuries, including meniscal injuries. Furthermore, dual-source spectral CT and CT arthrography are additional techniques that might be useful in the meniscal evaluation and visualize the integrity and position of the menisci [7].

The importance of timely diagnosis and repair of bucket handle meniscal tears cannot be overstated as it has a direct impact on long-term outcomes and the severity of the cartilage damage. The consequences of leaving these tears untreated can be severe, leading to knee instability, pain, and degenerative changes in the joint that raise the risk of developing chronic conditions, such as osteoarthritis [4].

However, by intervening early, the chances of re-establishing knee joint congruency, minimizing articular trauma, and preventing or reducing the burden of osteoarthritis in the future are significantly improved [8,9]. Prompt diagnosis and treatment can help to address the issue before it worsens and before the joint experiences further damage [8].

Early intervention not only reduces the likelihood of developing chronic conditions but also ensures that patients can return to their normal activities without the limitations caused by knee pain, instability, or restricted mobility. Therefore, it is essential for individuals who experience knee pain, swelling, or instability to seek medical attention promptly to avoid further damage and improve their quality of life. By working closely with their healthcare providers, individuals can develop a personalized treatment plan that is tailored to their specific needs and goals.

The diagnostic performance of CT scans for detecting dislocated bucket handle meniscal tears using MRI signs is still unclear. While some single case reports have presented the finding of a bucket handle meniscal tear on CT, the sensitivity and specificity of CT scans for this type of injury are not well established [10]. Most studies on the diagnostic accuracy of CT and MRI for meniscal injuries are from the 1980s and are based on outdated scanners [11,12,13]. A better understanding of the sensitivity and specificity of CT scans for detecting dislocated bucket handle meniscal tears could have significant implications for the evaluation and management of acute knee injuries, potentially reducing the need for more invasive diagnostic procedures and leading to earlier interventions and improved patient outcomes.

We hypothesized that CT is applicable to detect dislocated meniscal bucket handle tears. Therefore, the purpose of this study was to evaluate the performance and diagnostic accuracy of modern CT in detecting dislocated bucket handle meniscal tears in acute knee trauma with correlation to both MRI and arthroscopy.

## 2. Materials and Methods

### 2.1. Study Population

This retrospective single-center study was conducted according to the guidelines and recommendations of the Declaration of Helsinki with permission of the local ethics committee. The study period was from 1 December 2012 to 31 December 2021. All patients who underwent knee CT after acute knee trauma with following MRI (maximum time interval of six months) were eligible. Exclusion criteria were the presence of an acute fracture, the lack of both MRI and arthroscopy, and a patient’s age of less than 18 years. 

A total of ninety-six patients with acute knee trauma met the inclusion criteria and were enrolled in this study. The CT scan was performed on the day of the injury and the MRI imaging (internal or outside imaging) was acquired within a median of seven days after the accident. Furthermore, forty patients underwent arthroscopic correlation. Figure 1 is a flowchart of patient enrollment.

### 2.2. Image Acquisition 

All CT examinations were acquired on the same 3rd generation 128-slice single-source CT scanner (Somatom Definition Edge, Siemens Healthineers, Forchheim, Germany). For image acquisition, the institutional standard protocol with a tube voltage of 120 kV and a tube current of 90 mAs was used. The matrix size was set to 512 and the field of view was set to 146. Collimation was 0.6 × 128 mm, pitch factor 0.8, and gantry rotation time 1.0 s. Acquisitions were reconstructed with a slice thickness of 2 mm and an increment of 2 mm using standard filtered back projection. For osseous reconstructions, a sharp kernel (B75h) and for soft-tissue reconstructions a medium soft kernel (B30s) were chosen with a slice thickness of 2 mm and increment of 1 mm in axial, coronal, and sagittal orientation. Soft-tissue reconstructions in all three planes were additionally acquired in every CT scan after the adaptation of the institutional standard study protocol on 1 October 2018. Accordingly, only a portion of the 96 patients included since 2012 had an additional soft-tissue reconstruction (*n* = 45/96, 47%).

Due to common institutional circumstances, MR examinations were acquired at different radiological institutions and on different MR scanners from different vendors with different magnetic field strengths ranging between 1.5 T and 3.0 T. According to common practice and international guidelines, all study protocols consisted of coronal, sagittal, and transversal series with both T1-weighted and PD-respective T2-weighted images comprising different kinds of fat saturation techniques. Contrast-enhanced images were not considered for this study and were not evaluated.

### 2.3. Image Analysis 

CT data with axial, coronal, and sagittal bone kernel reconstructed images and if available with soft-tissue kernel reconstructions were rated pseudonymized and in randomized order independently by two radiologists with five and six years of experience in musculoskeletal radiology. To avoid recall bias in favor of CT images compared to MRI, CT imaging was analyzed first, followed by a memory washout interval of four weeks. To avoid recall bias in favor of CT images compared with MRI, CT images were first analyzed independently by each radiologist. In a separate session after four weeks, the MR images were evaluated in a consensus reading of both readers together. The images of CT and MRI were analyzed for the following established radiological signs of displaced bucket handle meniscal tears: double posterior cruciate ligament (PCL) sign, disproportional posterior horn sign, and double delta sign [14,15]. 

The reading scheme comprised the following items:Presence or absence (yes/no) of mentioned radiological signs of dislocated bucket handle meniscal tears (double delta sign, disproportional posterior horn sign, double PCL sign)Presence or absence (yes/no) of joint effusionDiagnostic confidence on a 5-point Likert scale [16]: 1 = “non-diagnostic imaging quality”, 2 = “slightly confident”, 3 = “fairly confident”, 4 = “confident”, and 5 = “very confident”.

Both readers were blinded to patients’ age as well as further patient information, such as accident course, referring clinician, health records, and radiological reports. The arthroscopic findings were taken from the surgical reports. All image analyses were performed at the same workstation with standardized conditions. 

### 2.4. Statistical Analysis 

Statistical analysis was performed using the software packages JMP (Version 15.2.0, SAS Institute, Cary, NC, USA) and SPSS (Version 28.0.0.0, IBM Corp., Armonk, NY, USA). Shapiro–Wilk-W-test was performed for continuous variables to assess normal distribution. Normally distributed variables are reported as arithmetic mean and standard deviation. Correlations of ordinal variables, such as the subjectively rated diagnostic confidence, were analyzed by likelihood ratio. *p*-values < 0.05 indicate statistical significance. 

Diagnostic quality criteria, such as sensitivity and specificity of the CT regarding the detection of bucket handle meniscal tears, were analyzed separately for both readers in an intra-observer comparison with MRI as the gold standard. Cohen’s Kappa was calculated for inter-observer agreement of the readers’ CT reading results. Values of 0.00–0.20 were considered as slight, 0.21–0.40 as fair, 0.41–0.60 as moderate, 0.61–0.80 as substantial, and 0.81–1.00 as almost perfect/perfect levels of agreement [17].

## 3. Results

### 3.1. Study Population 

The study cohort included 96 patients with CT imaging after an acute knee trauma with the absence of acute fractures (36 female, 60 male, 35.7 ± 12.2 years, Figure 1). All CT scans were performed on the day of the injury and the MRI images were acquired within a median of seven days after the accident. Furthermore, forty patients underwent arthroscopic correlation. Soft-tissue kernel reconstructions for CT were available in 45 cases (46.9%). 

According to MRI and arthroscopy, the prevalence of bucket handle meniscal tears was 28.1% (*n* = 27), with 55.6% (*n* = 15) located in the medial and 44.4% (*n* = 12) located in the lateral meniscus. In cases of present soft-tissue kernel reconstructions, the prevalence of dislocated bucket handle meniscal tears was 42.2% (*n* = 19).

### 3.2. Diagnostic Performance of CT Regarding Dislocated Bucket Handle Meniscal Tears 

MRI was available in all patients (100%) for gold-standard evaluation. Furthermore, 40 patients underwent subsequent arthroscopy for surgical confirmation. MRI diagnosis of a bucket handle meniscal tear was approved in all cases (100%). Of these 40 arthroscopic examinations, 37.5% (*n* = 15) cases had a dislocated bucket handle meniscal tear that was also identified on MRI and vice versa. In the remaining 62.5% (*n* = 25) bucket handle-negative arthroscopies, the MRI findings were also consistent with the arthroscopy.

Reader 1 detected 24 dislocated bucket handle meniscal tears on CT (sensitivity 88.9%, specificity 100%). Reader 2 detected 25 dislocated bucket handle meniscal tears on CT (sensitivity 92.6%, specificity 98.6%). When soft-tissue kernel reconstructions were available, both readers had a sensitivity of 94.7% and a specificity of 100%. Inter-reader agreement of readers 1 and 2 regarding the detection of dislocated bucket handle meniscal tears was 0.93 (Table 1).

The readers rated the radiological bucket handle signs on CT as follows (Figure 2, Figure 3 and Figure 4): double PCL sign (*n* = 22, sensitivity 81.5%, specificity 100%) and double delta sign (*n* = 1, sensitivity 3.7%, specificity 100%) were rated equally by both readers. The disproportional posterior horn sign was assessed slightly differently by reader 1 (*n* = 10, sensitivity 33.3%, specificity 98.6%) and by reader 2 (*n* = 12, sensitivity 37.0%, specificity 97.1%). In detail, reader 1 reported one false negative and reader 2 one false positive disproportional posterior horn sign.

The subjective diagnostic confidence regarding the assessment of dislocated bucket handle meniscal tears on CT was rated differently by the readers. It improved significantly (*p* < 0.0001) when soft-tissue kernel reconstructions were present from a median of 2/5 (‘slightly confident’) to a median of 4/5 (‘confident’) in both readers (see Table 1).

## 4. Discussion

The findings demonstrate an average sensitivity of 90.7% and specificity of 99.3% for the detection of bucket handle meniscal tears on CT by using the known MRI signs. 

Consistent with our results, Manco et al. could show a sensitivity of 88.5% and a specificity of 95.5% [12]. Considering additional soft-tissue reconstructions, the sensitivity in our study increased by 4 to 94.7% and the specificity increased to 100%. Furthermore, the inter-rater agreement increased to 1.0. MRI is currently the best imaging modality for the diagnosis of meniscal injuries, with a reported overall sensitivity of 93% [18,19].

Some approaches predict meniscal injuries in CT. Mui et al. and Chang et al. addressed the CT prediction of meniscal injuries in patients with tibial plateau fractures compared to MRI [20,21]. Unfortunately, the issue of sensitivity and specificity was not addressed in their study. 

The most established signs of bucket handle meniscal tears are the disproportional posterior horn sign, the double PCL sign, and the double delta sign. The disproportional posterior horn sign refers to the observation that the posterior horn of the meniscus appears more irregularly shaped than the anterior horn in imaging studies. The double PCL sign refers to the presence of two lines that extend from the intercondylar notch and resemble the PCL. The double delta sign is characterized by the presence of two triangular fragments that resemble a delta shape on sagittal MRI images. In addition to these established signs, there are many other reported radiological signs of bucket handle meniscal tears. Some of these signs may be synonymous or at least very similar to the established signs, such as the flipped fragment sign. Other reported signs include the truncated triangle sign, the bow tie sign, the ghost sign, the hidden fragment sign, and the fragment in notch sign [2,14,15,19,22,23,24].

Double PCL signs were found in our study on CT in 22 cases, with a sensitivity of 82% and specificity of 100%. The specificity of 100% is in line with multiple studies that have also shown this high specificity of 100% on MRI [14,15,24].

The disproportional posterior horn sign showed an average high specificity of 98% and a low sensitivity of 35%. Various authors have reported the sensitivity of disproportional posterior horn sign to be 21 to 28% on MRI, which is almost following our slightly higher sensitivity of 35% [3,15,25,26]. In agreement with our study, Vande Berg et al. reported an identically good specificity of 98% on MRI [26].

The double delta sign was only found in one case, with a respective sensitivity of 4% and a specificity of 100%. Dorsay et al. and Vande Berg et al. showed similar specificities on MRI of 90% and 98%, respectively [14,26]. Many studies provided varying sensitivities, ranging from 19 to 63% [3,14,15,25,26]. Vande Berg et al. reported a high variance of 12% of sensitivity in the same group of patients depending on the reader, indicating difficulty in diagnosing the double delta sign [26]. Considering the high variation in sensitivity in the literature of 44%, the sensitivity of 4% we found is within a reasonable range of the lowest reported sensitivity of 19% [26] and is not necessarily due to the imaging modality.

Nevertheless, many studies in the field of musculoskeletal radiology showed that different soft tissue pathologies can be assessed adequately not only by MRI but also by CT. For muscle tissue, Khil et al. showed that MRI and CT can be reliably used to measure fat content qualitatively and quantitatively in paraspinal back muscles [27]. Further, for other skeletal muscles, both CT and MRI revealed good and comparable interobserver reliability in quantifying abdominal skeletal muscle area [28]. CT is already making an important contribution to the assessment of disc herniation, especially in emergency care. Kim et al. found a comparable accuracy of CT and MRI in patients with suspected lumbar disc herniation [29]. This scenario is similar to the setting of the underlying study with acute knee trauma and the subsequent CT trauma scan of the knee in emergency care. Naraghi and White previously showed that CT is reserved for the diagnosis of suspected fractures and complex fractures, although associated ligamentous injuries may be evident on CT scans obtained for the evaluation of osseous injuries [30]. They demonstrated the CT-based diagnosis of a disrupted anterior cruciate ligament (ACL) with a soft-tissue window and the subsequent confirmation of a complete tear of the ACL by MRI.

A growing number of novel deep-learning-based and machine-learning-based algorithms offer exciting opportunities for improving diagnostic performances and a perspective on future goals and objectives in musculoskeletal radiology for various tasks [31,32]. Thus far, research mostly focuses on MRI rather than on CT imaging. Deep learning algorithms applied to the automatic segmentation of cartilage, meniscal, and ligamentous structures exist for MRI [33,34]. For CT, automatic segmentation tools based on deep learning are lacking in this area. However, deep learning algorithms already exist for other fields of musculoskeletal radiology, such as skeletal muscle segmentation, muscle atrophy, and the evaluation of pelvic muscles, fat, and bone for body composition assessment [35,36]. However, for example, deep-learning-based denoising algorithms, as already explored in cardiac imaging, may further improve CT image quality for the assessment of soft tissue pathologies in imaging in the future [37]. While CT is not typically the first-line imaging modality for the diagnosis of meniscal tears, it offers several advantages over MRI, including faster scan times, better visualization of bony structures, and the ability to perform three-dimensional reconstructions. Unfortunately, the use of CT for the diagnosis of meniscal tears has not been the subject of much research in recent years, with fewer studies focusing on the sensitivity and specificity of CT in detecting bucket handle meniscal tears [5,10,14].

The prevalence of bucket handle meniscal tears in our study was 28%, which is close to the literature, with 9–26% [3,18,38]. In contrast to the findings reported in the literature that bucket handle meniscal tears occur more frequently on the medial side, the approximately even distribution between medial and lateral bucket handle meniscal tears in the presented study population could be due to the fact that no classic ski region is located nearby [39,40]. 

This study has several limitations: (1) only grade 3 and 4 meniscal lesions were assessed and classified as meniscal tears [41]. Grade 1 and 2 lesions were not considered in our evaluation because these lesions are frequently in the older patients’ group and are not detectable on CT. (2) We have a wide age range within the patient population. Nevertheless, in our study, we found no correlation between the patient’s age, incidence, and number of bucket handle meniscal tears. (3) In our collective, the double delta sign occurred only once, which cannot be considered representative. (4) A further limitation is the retrospective and single-center study design. Of course, further prospective and multi-center approaches would be preferable to strengthen the results on this issue. Especially, investigations about the replicability of the study in a collective of junior radiologists or clinically working orthopedic surgeons will be interesting to strengthen our results. In addition, further developments based on software or artificial intelligence may improve the meniscoligamentous diagnostic performance of CT in the future.

Despite the reported results with high sensitivity and specificity of CT for the detection of dislocated bucket handle meniscal tears, the authors would like to emphasize that, in their opinion, CT is not auspicious for the detection of non-dislocated bucket handle meniscal tears or lower-grade meniscal lesions. MRI, of course, remains the preferred imaging modality for the assessment of meniscal lesions. Nevertheless, in cases of acute knee trauma when CT is clinically indicated, e.g., for the exclusion of a fracture, special attention on signs of a bucket handle meniscal tear seems worthwhile; additionally, soft-tissue reconstructions can be useful. A bucket handle meniscal tear detected on CT might improve therapy and patient outcome by accelerating or skipping an MRI evaluation to undergo surgery directly. This could help to triage patients, especially in institutions or regions with limited MRI capacity or long waiting times. 

## 5. Conclusions

This study demonstrates the high diagnostic sensitivity and specificity in detecting dislocated bucket handle meniscal tears of the knee in CT images after acute knee trauma, especially by using soft-tissue reconstructions. In the case of unavailable MRI, this may lead to earlier diagnosis, earlier therapy, and a reduction in consequential damage to injured knees in the future.

## Figures and Tables

**Figure 1 diagnostics-13-01295-f001:**
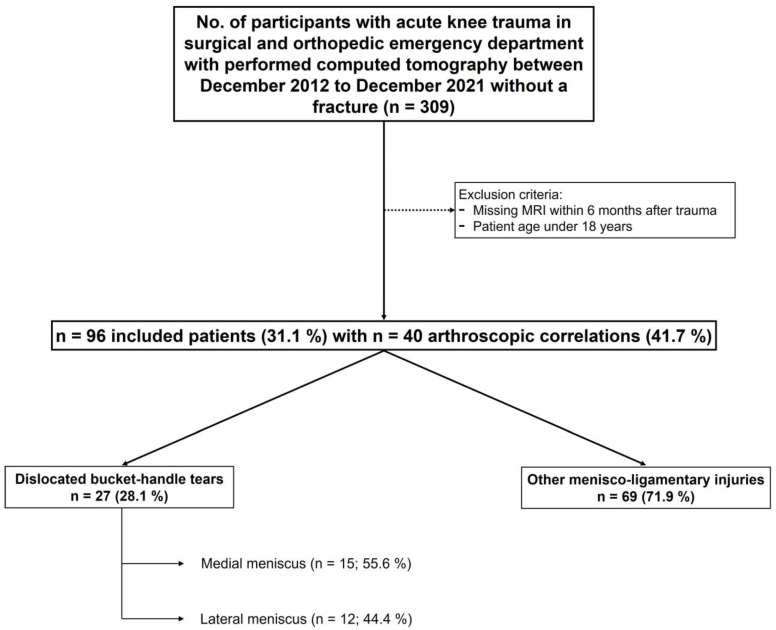
Flow chart of patient acquisition, inclusion, and exclusion criteria.

**Figure 2 diagnostics-13-01295-f002:**
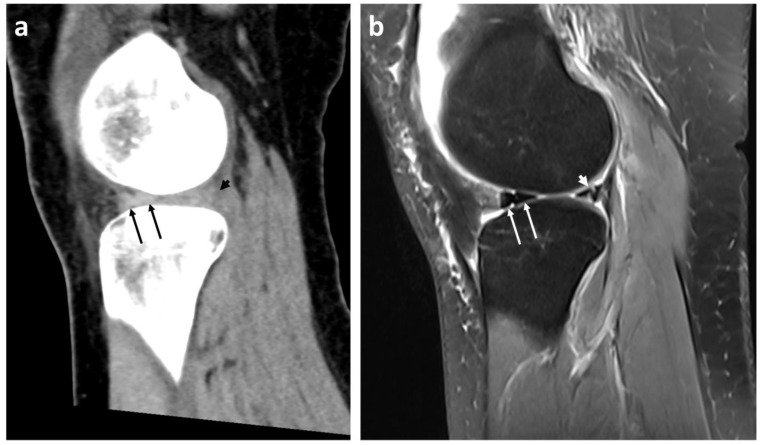
Double delta sign (arrows) and disproportional posterior horn sign (arrowheads) in a 27-year-old patient with a dislocated bucket handle tear of the lateral meniscus in the left knee on CT images in sagittal plane with soft-tissue kernel reconstruction (**a**) and MR images ((**b**), sagittal fat-suppressed intermediate-weighted Turbo Spin Echo).

**Figure 3 diagnostics-13-01295-f003:**
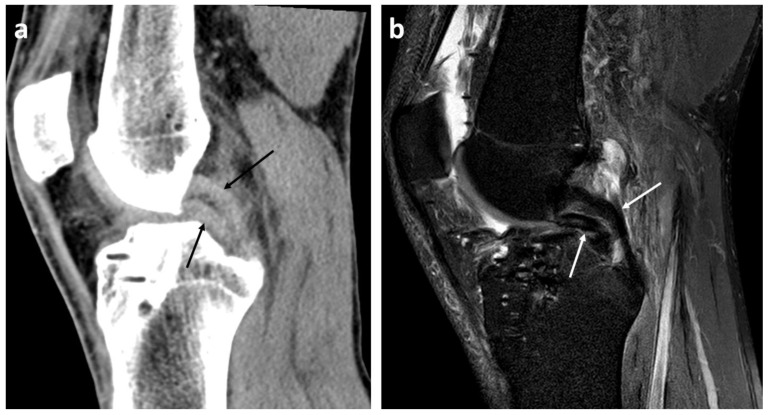
Double posterior cruciate ligament (PCL) sign (arrows) in the left knee in a 26-year-old patient with acute dislocated bucket handle meniscal tear of the medial meniscus on CT images with soft-tissue kernel reconstruction (**a**) and sagittal MR image ((**b**), PD SPIR TSE). PD = proton density weighted, TSE = Turbo Spin Echo, SPIR = Spectral Presaturation with Inversion Recovery.

**Figure 4 diagnostics-13-01295-f004:**
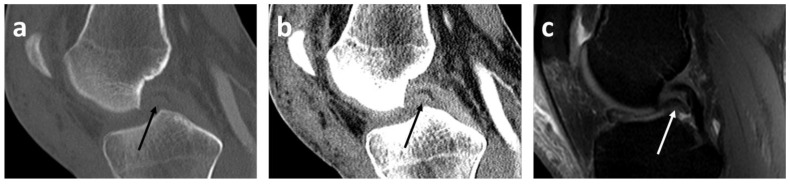
Double posterior cruciate ligament (PCL) sign (arrows) in the left knee on sagittal bone kernel CT scan with bone (**a**) and soft tissue windowing (**b**) as well as sagittal MRI scan ((**c**), PD SPAIR TSE) in a 49-year-old patient with a dislocated bucket handle meniscal tear of the medial meniscus. Note the improved distinguishability of the dislocated fragment in the CT images with soft tissue windowing (**b**) compared to bone windowing (**a**) on bone kernel reconstructions, which seems still little worse compared to soft-tissue kernel reconstructions (Figure 3). PD = proton density weighted, TSE = Turbo Spin Echo, SPAIR = Spectral Attenuated Inversion Recovery.

**Table 1 diagnostics-13-01295-t001:** The diagnostic performance for detecting dislocated bucket handle meniscal tears for all readers in all patients (*n* = 96). Of these 96 patients, additional soft-tissue reconstructions were present in 45 cases (47%). Diagnostic confidence was assessed using a 5-point Likert scale: 1 = “non-diagnostic imaging quality”, 2 = “slightly confident”, 3 = “fairly confident”, 4 = “confident”, and 5 = “very confident”. PPV: positive predictive value; NPV: negative predictive value.

Diagnostic Performance in CT					
Parameters		CT Images with Bone Kernel Reconstructions (*n* = 96)		CT Images with Soft-Tissue Kernel Reconstructions (*n* = 45)	
		Reader 1	Reader 2	Reader 1	Reader 2
Diagnostic confidence 5-point Likert scale	1 (*n*)	8	5	0	0
	2 (*n*)	36	32	4	2
	3 (*n*)	19	23	11	12
	4 (*n*)	27	31	21	24
	5 (*n*)	6	5	9	7
Sensitivity %		88.9	92.6	94.7	94.7
Specificity %		100	98.6	100	100
Accuracy %		96.9	96.9	97.8	97.8
PPV %		100	96.2	100	100
NPV %		95.8	97.1	96.3	96.3
False positive (*n*)		0	1	0	0
False negative (*n*)		3	2	1	1
True positive (*n*)		24	25	18	18
True negative (*n*)		69	68	26	26

## Data Availability

Data sharing not applicable.

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
