# Peer review of "Diagnostic Accuracy of 128-Slice Single-Source CT for the Detection of Dislocated Bucket Handle Meniscal Tears in the Setting of an Acute Knee Trauma—Correlation with MRI and Arthroscopy"

_diagnostics, 2023, doi:10.3390/diagnostics13071295_

Round 1

Reviewer 1 Report

Abstract

Row 17-18 does not make sense - please revise.

Row 29 - remove "the results suggest"

Introduction

A short sentence about the limitations of a CT scan should be also added

Methods

Please provide ethical board approval number and date.

Interesting to see that the percentage between medial and lateral meniscus dislocation is about 50-50% in your sample. The medial meniscus is usually in the 70-80% in other studies. Make sure you add a statement in the discussion section about this finding.

Please cite the Likert scale

Row 146 & others - prevalence measurement is usually in percentages. Therefore, I suggest using percentages first and add count number in brackets.

Results

Table 1. should be improved and given a more scientifically look. Removing colors will greatly improve it.

Please specify about interobserver and intraobserver statistical analysis. Both are important for these types of studies.

Discussion

The discussion section should always start with the main finding of your study and not with the aim or objectives.

The comparison with current literature is ok.

Conclusions

Authors should focus more on the study findings and present them in a short manner. The conclusion is too general.

References

62% of your references are older than 10 years. Please revise this.

Reviewer 2 Report

1.      The introduction is well-described, but please adding hypothesis of the study in last paragraph of introduction.

2.      About the methodology, why choice similar experience radiologists (5 and 6 year) to evaluate the CT? why not choice one junior and one senior radiologists? (In this way, it can show that CT can be applied by any experienced physician?)

3.      In table 1, why bone and soft-tissue reconstructions present different 96 and 45 patients?

4.      I wonder why not include 1 more orthopedic surgeon to evaluate the CT scan? Because one more clinician plus two radiologists to evaluate CT should be more convincing the result.

Round 2

Reviewer 2 Report

good revision for answering my queries